## Registered report

microbiology/health and disease and epidemiology/genomics

COVID-19, coronavirus SARS-CoV-2, diagnosis, sequence variation, polymerase chain reaction (PCR), primer–template mismatch

**Author for correspondence:**
Kashif Aziz Khan
e-mail: kakhan@yorku.ca

# Presence of mismatches between diagnostic PCR assays and coronavirus SARS-CoV-2 genome

## Kashif Aziz Khan and Peter Cheung

Department of Biology, York University, 4700 Keele Street, Toronto, Canada M3J 1P3

KAK, 0000-0003-3125-1497

Severe acute respiratory syndrome coronavirus 2 (SARS-CoV-2; initially named as 2019-nCoV) is responsible for the recent COVID-19 pandemic and polymerase chain reaction (PCR) is the current standard method for its diagnosis from patient samples. This study conducted a reassessment of published diagnostic PCR assays, including those recommended by the World Health Organization (WHO), through the evaluation of mismatches with publicly available viral sequences. An exhaustive evaluation of the sequence variability within the primer/probe target regions of the viral genome was performed using more than 17 000 viral sequences from around the world. The analysis showed the presence of mutations/mismatches in primer/probe binding regions of 7 assays out of 27 assays studied. A comprehensive bioinformatics approach for *in silico* inclusivity evaluation of PCR diagnostic assays of SARS-CoV-2 was validated using freely available software programs that can be applied to any diagnostic assay of choice. These findings provide potentially important information for clinicians, laboratory professionals and policy-makers.

## 1. Introduction

On 31 December 2019, a cluster of 41 pneumonia cases of unknown aetiology in Wuhan, China, were reported to the World Health Organization (WHO). Subsequently, a novel coronavirus of zoonotic origin, severe acute respiratory syndrome coronavirus 2 (SARS-CoV-2; initially named as 2019-nCoV), was isolated from the patients [1–3]. The virus has spread to more than 200 countries and territories resulting in global coronavirus disease 2019 (COVID-19) pandemic [4]. The rapid spread of the virus is partially attributed to the transmission by asymptomatic carriers or mildly symptomatic cases [5,6]. Early diagnostic testing is an important tool for policy-makers to make public health decisions to contain the outbreak.

The virus from the patients was identified and sequenced early in the outbreak [1,7] and resulted in the development of several polymerase chain reaction (PCR) detection protocols by multiple national organizations that were published by the WHO [8]. In addition, several other methods have been developed and published in the literature recently [5,7,9–15]. However, the molecular diagnosis of SARS-CoV-2 may be jeopardized by potential preanalytical and analytical vulnerabilities including lack of harmonization of primers and probes [16]. Given the potential for the viruses to mutate, genetic variations in the viral genome at primer/probe binding regions can result in potential mismatches and false-negative results [17]. For example, primer and template mismatches have been reported to impede proper diagnosis of several viruses including influenza virus [18–21], respiratory syncytial virus [22], dengue virus [23], rabies virus [24], human immunodeficiency virus-1 [25,26] and hepatitis B virus [27,28].

SARS-CoV-2 is an enveloped positive-strand RNA virus classified as a member of family *Coronaviridae* in the genus *Betacoronavirus* along with SARS-CoV and Middle East respiratory syndrome (MERS)-CoV [29]. The sequence analysis of SARS-CoV-2 isolates has shown that its single-stranded RNA genome is approximately 30 kb in size [1,7,30]. Based on similarity with SARS-CoV, SARS-CoV-2 genome has been predicted to encode at least 10 open reading frames (ORFs) for structural and accessory proteins. As per current annotation (NC_045512.2), these viral ORFs encode replicase ORF1ab, spike (S), envelope (E), membrane (M) and nucleocapsid (N), and at least six accessory proteins (3a, 6, 7a, 7b, 8 and 10) [31].

Human coronaviruses encode a proofreading exoribonuclease, nsp14-ExoN, for maintaining replication fidelity and thus have a relatively slower mutation rate than other RNA viruses [32,33]. SARS-CoV-2 encodes nsp14-ExoN as well [1], but mutations have been described in the genome for circulating SARS-CoV-2 [34–38]. Some laboratories have performed the alignment of diagnostic primers/probes with a limited number of viral sequences and have reported some mismatches [39,40] which may lead to false-negative results [41]. The use of several commercially developed diagnostic assays has also been permitted around the world with limited regulatory approval due to the pandemic emergency [42]. However, the limit of detection of these assays differs considerably and can also lead to false-negative results [43]. As there are already reports of false-negative diagnosis of COVID-19 [44–48], there is a need for verification of potential primer/probe mismatch with the sequences of viral isolates being isolated from around the world. The American Society for Microbiology COVID-19 International Summit held on 23 March 2020 recommended routine verification of sequence mutations in primer and probe binding regions of the viral genome for optimal virus detection [49].

The objective of this study is the *in silico* reassessment of previously published PCR primers/probes for COVID-19 diagnosis. This was performed through the evaluation of the sequence variability within the primer/probe target regions of SARS-CoV-2 viral isolates from around the world. The absence of any mutations and mismatches in target regions of the assay used would provide a higher degree of confidence in the test results obtained while the presence of mutations could help guide the strategies for the reassessment of diagnostic assays. We believe that these findings provide potentially important information for clinicians, laboratory professionals and policy-makers.

# 2. Methods

This study was pre-registered on the Open Science Framework (OSF); the accepted Stage 1 registration can be viewed at (https://osf.io/ym8gc). Minor deviations from protocol are identified in footnotes. The study design planner is included in table 1. The summary of the sequence tracing pipeline is shown in figure 1.

## 2.1. Selection of primers and probes

A total of 27 PCR primer-probe sets were selected based on literature review [9,10,12–15,50–52] and on the assays posted on WHO website [8] originally developed by seven different national institutions including Chinese Center for Disease Control and Prevention (China CDC), China; Institut Pasteur, Paris, France; US Centers for Disease Control and Prevention (CDC), USA; National Institute of Infectious Diseases, Japan; Charité – Universitätsmedizin Berlin Institute of Virology, Germany; The University of Hong Kong, Hong Kong; and National Institute of Health, Thailand.

**Table 1.** Study design planner.

| question | hypothesis | sampling plan (e.g. power analysis) | analysis plan | interpretation given different outcomes | obtained results and interpretation |
|---|---|---|---|---|---|
| are there any mutations in the primer/probe binding regions of the SARS-CoV-2 genome for PCR assays published in the literature? | as the virus can potentially mutate during the outbreak, mutations in the primer/probe binding regions can result in mismatches with primer/probe template | 17 026 viral isolates would be downloaded from GISAID EpiCov database<br><br>*inclusion criteria:*<br>only full length (>29 000 bp)<br><br>*exclusion criteria:*<br>the sequences with stretches of NNNs, ambiguous sequences, and missing sequences in the region of interest (ROI) will be considered low quality and would be excluded | sequences would be aligned using MAFFT<br>low-quality sequences would be excluded from the alignment and sequence variability would be traced *in silico* using SequenceTracer<br>the highly variable region, if any, would be further analysed for nucleotide composition at each position using positional nucleotide numerical summary (PNNS)<br>the complete genome of Wuhan-Hu-1 from the National Center for Biotechnology Information (NCBI) would act as a positive control (NCBI Reference Sequence: NC_045512.2) | in the event of a negative result, it would be concluded that there is no evidence of a difference between primer/probe and viral isolates<br>this would serve as a reference for researchers and laboratory professionals using PCR assays for the detection of SARS-CoV-2 | the analysis showed the presence of mismatches/mutations in primer/probe binding regions of 7 assays out of 27 assays studied |

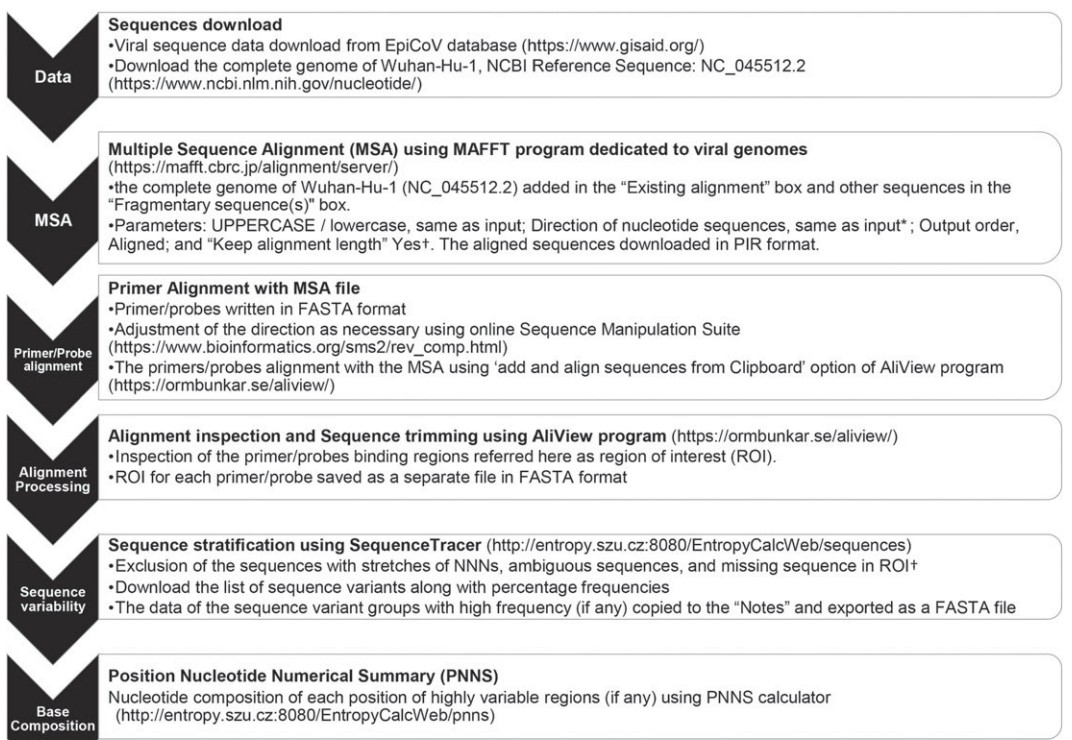

**Figure 1.** Sequence tracing pipeline used in the study. *The direction can be adjusted by selecting the option 'Adjust direction according to the first sequence', if needed. †The change was made with editorial approval after Stage 1.

## 2.2. Sequencing data

The complete genome sequences of the virus were downloaded from the Global Initiative on Sharing All Influenza Data (GISAID) EpiCoV database [53]. As of 7 May 2020, it hosted a total of 17 175 SARS-CoV-2 sequences isolated from humans. By applying the complete genome (greater than 29 000 bp) filter, a total of 17 026 sequences were included in the study that are available upon free registration (https://www.gisaid.org/). SARS-CoV-2 is an RNA virus, but the data are shown in DNA format as per scientific convention. The sequences are shared by the laboratories around the world and a list of accession numbers is included in electronic supplementary material, file S1. It is recognized that this study is not immune to the geographical bias present in academic and scientific research. As the data were sampled from a global sequence database, it is possible that data may originate from high-income countries like the literature in other disciplines [54,55]. In addition, it is possible that data from certain countries or regions are excluded based on the exclusion criteria of low-quality data that may skew the data geographically. Another reason for possible data skew may be the origin of the current pandemic being China. Indeed, a recent study analysed the publications in COVID-19 literature hub LitCovid [56] and observed that more than 30% of articles were related to China [57]. These aspects of possible bias and data skew are addressed in the Discussion to make sure that the valid conclusions are drawn from the data in terms of geographical correlation.

## 2.3. Multiple sequence alignment and alignment processing

Multiple sequence alignment (MSA) was performed using MAFFT (Multiple Alignment with Fast Fourier Transform) program v. 7 dedicated to closely related viral genomes [58,59] available online (https://mafft.cbrc.jp/alignment/server/). The complete genome of Wuhan-Hu-1 downloaded from NCBI on 7 May 2020 was included as a reference, which is 29 903 bp long (NCBI Reference Sequence: NC_045512.2). The aligned sequences were downloaded in PIR format. Each primer/probe was aligned with the MSA and the binding region referred to here as region of interest (ROI) was inspected using the AliView program 1.26 [60]. To evaluate the sequence variability in target regions of previously published primers/probes, the ROI for each primer/probe set was saved as a separate file in FASTA format.

## 2.4. Sequence variation in primer/probe binding regions in SARS-CoV-2 genome

The MSA sequence for forward primer, probe and reverse primer were stratified using the SequenceTracer module (http://entropy.szu.cz:8080/EntropyCalcWeb/sequences) of the Alignment Explorer [61]. This tool segregated sequences into discrete groups of identical sequence variants along with their frequency for each primer/probe. The sequences with stretches of NNNs, ambiguous sequences in ROI and missing sequences[1] were excluded from the study. Subsequently, a threshold[2] (0.5% of all sequences included) was defined to remove extremely low prevalent variants and sequencing errors in the data as described previously [61]. Thus, only the sequence variants with at least 0.5% incidence were further considered. The viral isolates were reported as the frequency of hits with perfect primer match and hits with mismatches along with a summary of mutated nucleotides for each primer/probe. The distribution of the sequence variants in three primers/probes with the highest frequency of mismatches were analysed geographically. As the sequence variation was moderate, the base composition of each nucleotide position was not analysed. As noted in the registered Stage 1 protocol (https://osf.io/ym8gc), this analysis can be performed using the positional nucleotide numerical summary (PNNS) calculator (http://entropy.szu.cz:8080/EntropyCalcWeb/pnns) of the Alignment Explorer [61].

## 3. Results

The sequence tracing pipeline (figure 1) was applied to the comprehensive sequence dataset of 17 027 SARS-CoV-2 sequences for each PCR primer/probe. To determine the sequence variability in the primer/probe binding regions, all the sequences in the dataset were aligned using MAFFT. Next, for each PCR assay, the MSA file was trimmed to include only the primer or probe binding regions referred to here as ROI. The sequence file for each primer/probe was submitted to SequenceTracer to segregate into discrete groups of identical sequence variants and presented a detailed view of the nucleotide variation in each ROI along with the frequency of each variant (figures 2 and 3; electronic supplementary material, file S2). All the sequences showing ambiguous sequences were grouped as 'outgroup1', short sequences were grouped as 'outgroup2' and missing sequences were grouped as 'excluded'. These three groups were not included in the analysis (collectively referred here as 'removed'), and the number of 'informative' sequences was calculated by subtracting these three groups from the total number of sequences. The informative group was then divided into hits with a perfect match and hits with mismatches for each primer and probe (table 2). It is not surprising that most primer/probe binding regions show mutations/mismatches with at least a couple of sequences but some of those may be extremely low prevalent variants and sequencing errors in the data. To minimize the effect of such sequences on the analysis, a threshold of 0.5% was then defined where only the sequence variants with at least 0.5% incidence were further considered as described previously [61]. The frequency of the sequences with the perfect match and with mismatches was then calculated from sequences above the threshold for each primer and probe. The summary of the analysis for 27 assays is presented in table 2.

It was observed that the primers/probe of 20 assays out of 27 assays tested showed a perfect match with the template at the defined threshold (table 2). It was further observed that the forward primer of CN-CDC-N showed three nucleotide mismatches with 18.8% of viral sequences (table 3 and figure 2*a*). In addition, the US-CDC-N-1 probe and the US-CDC-N-3 forward primer showed one mismatch with 1.6% and 1.2% viral sequences, respectively (table 3 and figure 3). The reverse primer of NIID-JP-N also showed one mismatch with all the sequences (table 3; electronic supplementary material, file S2). The probe of Chan-ORF1ab showed one mismatch with 0.9% of sequences while one mismatch in the reverse primer for all the sequences (table 3; electronic supplementary material, file S2). One mismatch was also observed with all the sequences for the probe of Young-N (table 3; electronic supplementary material, file S2). Most of the mismatches observed were not near the 3′ end of primers but some were in the probe binding regions. Many diagnostic assays have included degenerate nucleotides to increase the inclusivity of the assay for SARS-CoV and bat-SARS-related CoVs, but in certain cases, this is even detrimental for inclusive detection of SARS-CoV-2. For example, the Charité-ORF1b

---

[1]SequenceTracer removes the missing sequences in ROI. The exclusion criterion of missing sequences was clarified with editorial approval after Stage 1 acceptance and prior to observation of the data.

[2]The threshold was decided before Stage 1 acceptance. However, it was not clearly mentioned in the Stage 1 protocol and a previous study was referenced only.

**Figure 2.** Sequence variants in primers and probe binding regions for CN-CDC-N (*a*) and Charité-ORF1b (*b*): sequence variants in 17 026 viral genome sequences aligned to the primer/probe binding regions (5′ → 3′) along with the number of sequence variants and the frequency of each variant in descending order. The dots indicate an identical nucleotide. The horizontal double bar indicates the threshold (greater than or equal to 0.5%). The binding region of reverse primer is reverse complemented. As an example, the removed and informative sequences are indicated with vertical bars. outgroup1, ambiguous sequences; outgroup2, short sequences.

reverse primer contains an S (G or C) but all the viral sequences (in total 17 002) contain a T at this position (table 3 and figure 2*b*). Some of the other mutations observed in the primer/probe binding regions that did not pass the defined threshold include T13402G, C15540T, A28338G, C28846T, C28887T, C28896G, C29144T, T29148C and A29188T. Some of these are near the 3′ end of primers (figures 2 and 3; electronic supplementary material, file S2).

The majority of the sequences included in this study originated from Europe (9410) and North America (4759), while there were only 136 sequences from Africa, 7 from Central America and 142 from South America. The UK and the USA were among the countries with the highest number of sequences included (figure 4*a*; electronic supplementary material, file S3). The geographical distribution of the CN-CDC-N forward primer, US-CDC-N-1 probe and US-CDC-N-3 forward primer mismatches showed that it is distributed globally. However, mismatches with the CN-CDC-N forward primer were mostly found in Europe, while mismatches with the US-CDC-N-1 probe and the US-CDC-N-3 forward primer were found mostly in Australia and Asia (figure 4; electronic supplementary material, file S3).

## 4. Discussion

This study exhaustively evaluated the genetic diversity in the primer/probe binding regions of 27 previously published SARS-CoV-2 diagnostic assays including those recommended by WHO. The data presented in this study show mismatches in seven assays, highlighting the need for keeping the assay current through regular verification of sequence variation in PCR primer/probe binding regions. The other 20 assays show a perfect match with 100% of sequences at the defined threshold of 0.5%. This observation is in line with the estimates of the moderate mutation rate in the SARS-CoV-2 genome similar to the SARS-CoV genome [63,64]. It has been estimated that the mutation rate in the genome of coronaviruses is less than other RNA viruses while much higher than DNA viruses and the host [65,66]. Although all the sequences with mismatches were grouped in comparison to sequences with a perfect match, not all mismatches necessarily result in false-negative results. The

## (a) US-CDC-N-1

**Forward** — GACCCCAAAATCAGCGAAAT

| Group Number | Variant Count | Frequency % | |
|---|---|---|---|
| 1 | 16970 | 99.665 | .................... |
| 2 | 6 | 0.035 | ...........t...... |
| 3 | 3 | 0.018 | ....t.............. |
| 4 | 2 | 0.012 | ................c.. |
| 5 | 1 | 0.006 | ...t.............. |
| 6 | 1 | 0.006 | t................. |
| 7 | 1 | 0.006 | ........c........ |
| 8 | 1 | 0.006 | ..a.............. |
| 9 | 1 | 0.006 | ...t............. |
| 10 | 1 | 0.006 | ................g. |
| outgroup1 | 40 | 0.235 | |

**probe** — ACCCCGCATTACGTTTGGTGGACC

| Group Number | Variant Count | Frequency % | |
|---|---|---|---|
| 1 | 16647 | 97.768 | .................... |
| 2 | 273 | 1.603 | ..t................. |
| 3 | 7 | 0.041 | ......................t. |
| 4 | 6 | 0.035 | ...................t... |
| 5 | 4 | 0.023 | .a.................. |
| 6 | 3 | 0.018 | .t.................. |
| 7 | 3 | 0.018 | ......t............. |
| 8 | 2 | 0.012 | ...g................ |
| 9 | 2 | 0.012 | ...........t........ |
| 10 | 2 | 0.012 | ...................t |
| 11 | 2 | 0.012 | ...........t........ |
| 12 | 1 | 0.006 | ...............t.... |
| 13 | 1 | 0.006 | ...t................ |
| 14 | 1 | 0.006 | ..................c.... |
| 15 | 1 | 0.006 | ...............a.... |
| outgroup1 | 71 | 0.417 | |
| outgroup2 | 1 | 0.006 | |

**reverse** — CAGATTCAACTGGCAGTAACCAGA

| Group Number | Variant Count | Frequency % | |
|---|---|---|---|
| 1 | 16876 | 99.113 | ..................... |
| 2 | 58 | 0.341 | ...g................. |
| 3 | 25 | 0.147 | ........a........... |
| 4 | 6 | 0.035 | ...................g... |
| 5 | 4 | 0.023 | ........t........... |
| 6 | 2 | 0.012 | ..t................. |
| 7 | 1 | 0.006 | ....................g.. |
| 8 | 1 | 0.006 | .................t..... |
| 9 | 1 | 0.006 | ...............t...... |
| 10 | 1 | 0.006 | ..................t. |
| 11 | 1 | 0.006 | ...............t...... |
| 12 | 1 | 0.006 | ....c............... |
| 13 | 1 | 0.006 | ...................c. |
| 14 | 1 | 0.006 | .....a.............. |
| outgroup1 | 48 | 0.282 | |

## (b) US-CDC-N-3

**Forward** — GGGAGCCTTGAATACACCAAAA

| Group Number | Variant Count | Frequency % | |
|---|---|---|---|
| 1 | 16747 | 98.356 | .................. |
| 2 | 196 | 1.151 | .......c.......... |
| 3 | 13 | 0.076 | .........t........ |
| 4 | 3 | 0.018 | ...........c...... |
| 5 | 2 | 0.012 | ..........c....... |
| 6 | 1 | 0.006 | ..............t... |
| 7 | 1 | 0.006 | ..........g....... |
| 8 | 1 | 0.006 | .......c........t..... |
| outgroup1 | 62 | 0.364 | |
| outgroup2 | 1 | 0.006 | |

**probe** — AYCACATTGGCACCCGCAATCCTG (Y=C/T)

| Group Number | Variant Count | Frequency % | |
|---|---|---|---|
| 1 | 16922 | 99.383 | .T................. |
| 2 | 20 | 0.117 | .t.................t.. |
| 3 | 16 | 0.094 | .t..t............. |
| 4 | 9 | 0.053 | .t...........t.... |
| 5 | 7 | 0.041 | .t................t |
| 6 | 3 | 0.018 | .tt............... |
| 7 | 2 | 0.012 | .t.............t... |
| 8 | 2 | 0.012 | .t......t......... |
| 9 | 2 | 0.012 | .t..t............. |
| 10 | 1 | 0.006 | .t........t....... |
| 11 | 1 | 0.006 | .g................ |
| 12 | 1 | 0.006 | .t.................a.. |
| outgroup1 | 41 | 0.241 | |

**reverse** — CAATGCTGCAATCGTGCTACA

| Group Number | Variant Count | Frequency % | |
|---|---|---|---|
| 1 | 16952 | 99.560 | .................... |
| 2 | 27 | 0.159 | .......t.......... |
| 3 | 6 | 0.035 | .....t............ |
| 4 | 3 | 0.018 | ............t..... |
| 5 | 1 | 0.006 | .t................ |
| 6 | 1 | 0.006 | ..........a....... |
| 7 | 1 | 0.006 | t................. |
| 8 | 1 | 0.006 | ..............t.... |
| 9 | 1 | 0.006 | ..........c....... |
| outgroup1 | 34 | 0.200 | |

**Figure 3.** Sequence variants in primers and probe binding regions for US-CDC-N-1 (*a*) and US-CDC-N-3 (*b*): sequence variants in 17 026 viral genome sequences aligned to the primer/probe binding regions ($5' \rightarrow 3'$) along with the number of sequence variants and the frequency of each variant in descending order. The dots indicate an identical nucleotide. The horizontal double bar indicates the threshold (greater than or equal to 0.5%). The binding region of reverse primer is reverse complemented. outgroup1, ambiguous sequences; outgroup2, short sequences; excluded.

effects of mismatch between primers/probes and template depend upon position and number of mismatches. Most of the mismatches observed in primers of SARS-CoV-2 diagnostic assays were not near the 3′ end and may be tolerated. Mismatches at the 3′ end are known for their deleterious effect on PCR amplification [17,67,68], but single mismatches, especially more than 5 bp far from the 3′ end, have a moderate effect on PCR amplification and are unlikely to significantly affect the assay performance [67]. Three assays showed a single nucleotide mismatch in the probe binding region. PCR amplification is more prone to mismatches in the probe region and even a single mismatch may reduce the sensitivity of the assay and lead to false-negative results due to the prevention of probe binding and subsequence fluorescence [22,28,69–71]. In the scenarios where mismatches were tolerated, one additional mutation resulted in reduced RT-qPCR sensitivity for the detection of influenza A virus [18].

Despite the ability of single mismatches to be tolerated, it is important to consider that mismatches need to be corrected if found in most of the viral sequences available. For example, the reverse primer of Charité-ORF1b shows a mismatch with all the viral sequences (a total of 17 002). This mismatch has also been observed in 990 viral sequences along with the lower sensitivity of this assay in a recent preprint [72]. Similarly, the NIID-JP-N reverse primer also shows a mismatch with all the sequences. This assay released by WHO was subsequently corrected by the authors in a separate study [51]. Although they show no difference in the performance of both assays, there is no apparent reason for not correcting the mismatch in the primer. The WHO recommended assays of SARS-CoV-2 were developed by multiple national organizations early in the outbreak with limited genomic sequence data available and have been instrumental for the diagnosis of COVID-19. However, some of the assays have not been reassessed in the light of the risk of mutations during viral evolution. Based on the analysis of 17 027 viral sequences, this study demonstrates the presence of mutations/mismatches in the primer/probe binding regions of some published assays (table 3). Sequences adjustments to these primers/probes need to be assessed experimentally using viral strains or nucleic acid coupled with subsequent experimental performance using clinical samples. With increasing concern of false-negative COVID-19 diagnosis and poor sensitivity of diagnostic PCR in certain cases [73,74],

**Table 2.** Reassesment of 27 published PCR diagnostic assays using 17 027 SARS-CoV-2 genome sequences.

| gene target | assay name[a] | country | F/P/R[b] | sequence (5'–3') | position[c] | total number of sequences | | | | | above threshold (≥0.5%)[d] | | reference(s) |
|---|---|---|---|---|---|---|---|---|---|---|---|---|---|
| | | | | | | removed | informative | perfect match | with mismatches | total | perfect match (%) | with mismatches (%) | |
| ORF1ab | Yip-ORF1ab | China | F | ATGCATTTGCATCAGAGGCT | 1866->1885 | 85 | 16 942 | 16 911 | 31 | 16 911 | 100 | | [14] |
| | | | R | TTGTTATAGGGCCTTCTGT | 1970<-1951 | 168 | 16 859 | 16 855 | 4 | 16 855 | 100 | | |
| | Pasteur-ORF1ab-1 | France | F | ATGAGCTTAGTCCTGTTG | 12 690->12 707 | 54 | 16 973 | 16 973 | 0 | 16 973 | 100 | | [8] |
| | | | P | AGATGTCTTGTGCTGCGGGTA | 12 717->12 737 | 25 | 17 002 | 16 997 | 5 | 16 997 | 100 | | |
| | | | R | CTCCCTTTGTTGTTGTTGT | 12 797<-12 780 | 28 | 16 999 | 16 945 | 54 | 16 945 | 100 | | |
| | Pasteur-ORF1ab-2 | France | F | GGTAACTGGTATGATTTCG | 14 080->14 098 | 46 | 16 981 | 16 981 | 0 | 16 981 | 100 | | [8] |
| | | | P | TCATACAAACACGGCCAGG | 14 105->14 123 | 45 | 16 982 | 16 958 | 24 | 16 958 | 100 | | |
| | | | R | CTGGTCAAGGTTAATATAGG | 14 186<-14 167 | 50 | 16 977 | 16 939 | 38 | 16 939 | 100 | | |
| | CN-CDC-ORF1ab | China | F | CCCTGTGGGTTTTACACTTAA | 13 342->13 362 | 40 | 16 987 | 16 977 | 10 | 16 977 | 100 | | [8,12] |
| | | | P | CCGTCTGCGGTATGTGGAAAGGTTATGG | 13 377->13 404 | 1059 | 15 968 | 15 905 | 63 | 15 905 | 100 | | |
| | | | R | ACGATTGTGCATCAGCTGA | 13 460<-13 442 | 1037 | 15 990 | 15 978 | 12 | 15 978 | 100 | | |
| | Young-ORF1ab | Singapore | F | TCATTGTTAATGCCTATATTAACC | 14 155->14 178 | 51 | 16 976 | 16 969 | 7 | 16 969 | 100 | | [15] |
| | | | P | AACTGCGAGTGCACATGTTGACA | 14 193->14 215 | 67 | 16 960 | 16 939 | 21 | 16 939 | 100 | | |
| | | | R | CACTTAATGTAAGGCTTTGGTTAAG | 14 243<-14 220 | 25 | 17 002 | 16 983 | 19 | 16 983 | 100 | | |
| | Charité-ORF1b | Germany | F | GTGARATGGTCATGTGTGGCGG | 15 431->15 452 | 71 | 16 956 | 16 908 | 48 | 16 908 | 100 | | [8,50] |
| | | | P | CAGGTGGAACCTCATCAGGAGATGC | 15 470->15 494 | 43 | 16 984 | 16 976 | 8 | 16 976 | 100 | | |
| | | | R | CARATGTTAAASACACTATTAGCATA | 15 530<-15 505 | 23 | 17 004 | 0 | 17 004 | 17 002 | 0.0 | 100 | |
| | Won-ORF1ab | South Korea | F | CATGTGTGGCGGTTCACTAT | 15 441->15 460 | 45 | 16 982 | 16 972 | 10 | 16 972 | 100 | | [13] |
| | | | R | TGCATTAACATTGGCCGTGA | 15 558<-15 539 | 29 | 16 998 | 16 931 | 67 | 16 931 | 100 | | |
| | Chan-ORF1ab | China | F | CGGATACAGTCTTRCAGGCT | 16 220->16 239 | 69 | 16 958 | 16 946 | 12 | 16 946 | 100 | | [9] |
| | | | P | TTAAGATGTGGTGCTTGCATACGTAGAC | 16 276->16 303 | 84 | 16 943 | 16 786 | 157 | 16 930 | 99.1 | 0.9 | |
| | | | R | GTGTGATGTTGAWATGACATGGTC | 16 353<-16 330 | 86 | 16 941 | 0 | 16 941 | 16 932 | 0.0 | 100 | |
| | HKU-ORF1b | Hong Kong | F | TGGGGYTTTACKRGGTAACCT | 18 778->18 797 | 61 | 16 966 | 16 932 | 34 | 16 932 | 100 | | [8,52] |
| | | | P | TAGTTGTGATGCWATCATGACTAG | 18 849->18 872 | 41 | 16 986 | 16 976 | 10 | 16 976 | 100 | | |
| | | | R | AACRCGCTTAACAAAGCACTC | 18 909<-18 889 | 48 | 16 979 | 16 958 | 21 | 16 958 | 100 | | |

(Continued.)

**Table 2.** (*Continued*.)

| gene target | assay name[a] | country | F/P/R[b] | sequence (5'→3') | position[c] | removed | informative | perfect match | with mismatches | total | perfect match (%) | with mismatches (%) | reference(s) |
|---|---|---|---|---|---|---|---|---|---|---|---|---|---|
| S | Young-S | Singapore | F | TATACATGTCTCTGGGACCA | 21 763–>21 782 | 91 | 16 936 | 16 907 | 29 | 16 907 | 100 | | [15] |
| | | | P | CTAAGAGGTTTGATAACCCTGTCGTCACC | 21 789–>21 816 | 90 | 16 937 | 16 910 | 27 | 16 910 | 100 | | |
| | | | R | ATCCAGCCTCTTATTATGTTAGAC | 21 876<–21 853 | 99 | 16 928 | 16 907 | 21 | 16 907 | 100 | | |
| | Chan-S | China | F | CCTACTAAATTAAATGATCTCTGCTTTACT | 22 712–>22 741 | 254 | 16 773 | 16 768 | 5 | 16 768 | 100 | | [9] |
| | | | P | CGCTCCAGGGGAAACTGGAAAG | 22 792–>22 813 | 262 | 16 765 | 16 752 | 13 | 16 752 | 100 | | |
| | | | R | CAAGCTATAACGGAGCCTGTGA | 22 869<–22 849 | 65 | 16 962 | 16 956 | 6 | 16 956 | 100 | | |
| | Won-S | South Korea | F | CTACATGCACAGCAACTGT | 23 114–>23 133 | 872 | 16 155 | 16 126 | 29 | 16 126 | 100 | | [13] |
| | | | R | CACCTGTGCCTGTTAAACCA | 23 213<–23 194 | 29 | 16 998 | 16 987 | 11 | 16 987 | 100 | | |
| E | Won-E | South Korea | F | TTCGGAAGAGACAGGTACGTT | 26 259–>26 279 | 33 | 16 994 | 16 986 | 8 | 16 986 | 100 | | [13] |
| | | | R | CACACAATCGATGCGCAGTA | 26 365<–26 346 | 83 | 16 944 | 16 938 | 6 | 16 938 | 100 | | |
| | Charité-E | Germany | F | ACAGGTACGTTAATAGTTAATAGCGT | 26 269–>26 294 | 47 | 16 980 | 16 975 | 5 | 16 975 | 100 | | [8,50] |
| | | | P | ACACTAGCCATCCTTACTGCGCTTCG | 26 332–>26 357 | 75 | 16 952 | 16 928 | 24 | 16 928 | 100 | | |
| | | | R | ATATTGCAGCAGTACGCACACA | 26 381<–26 360 | 89 | 16 938 | 16 928 | 10 | 16 928 | 100 | | |
| | Huang-E | China | F | ACTTCTTTTTCGCTTTCGTGGT | 26 295–>26 318 | 80 | 16 947 | 16 925 | 22 | 16 925 | 100 | | [10] |
| | | | P | CTAGTTACACTAGCCATCCTTACTGC | 26 326–>26 351 | 81 | 16 946 | 16 920 | 26 | 16 920 | 100 | | |
| | | | R | GCAGGAGTAGGCACACAATC | 26 376<–26 357 | 90 | 16 937 | 16 928 | 9 | 16 928 | 100 | | |
| | Niu-E | China | F | TTCTTGCTTTCGTGGTATTC | 26 303–>26 322 | 78 | 16 949 | 16 926 | 23 | 16 926 | 100 | | [12] |
| | | | P | GTTACACTAGCCATCCTTACTGCGCTTCGA | 26 329–>26 358 | 82 | 16 945 | 16 921 | 24 | 16 921 | 100 | | |
| | | | R | CACGTTAACAATATTGCAGC | 26 391<–26 372 | 111 | 16 916 | 16 911 | 5 | 16 911 | 100 | | |

(*Continued.*)

**Table 2.** (*Continued.*)

| gene target | assay name[a] | country | F/P/R[b] | sequence (5′–3′) | position[c] | total number of sequences | | | | above threshold (≥0.5%)[d] | | | reference(s) |
| | | | | | | removed | informative | perfect match | with mismatches | total | perfect match (%) | with mismatches (%) | |
|---|---|---|---|---|---|---|---|---|---|---|---|---|---|
| N | CN-CDC-N | China | F | GGGGAACTCCTCCTGCTAGAAT | 28 881–>28 902 | 170 | 16 857 | 13 533 | 3324 | 16 662 | 81.2 | 18.8 | [8,12] |
| | | | P | TTGCTGCTGCTTGACAGATT | 28 934–>28 953 | 85 | 16 942 | 16 939 | 3 | 16 939 | 100 | | |
| | | | R | CAGACGATTTGCTCGTCAAGCTG | 28 979<–28 958 | 92 | 16 935 | 16 905 | 30 | 16 905 | 100 | | |
| | NIH-TH_N | Thailand | F | CGTTTGGTGGACCCTCAGAT | 28 320–>28 339 | 52 | 16 975 | 16 893 | 82 | 16 893 | 100 | | [8] |
| | | | P | CAACTGGCAGTAACCA | 28 341–>28 356 | 42 | 16 985 | 16 946 | 39 | 16 946 | 100 | | |
| | | | R | CCCCACTGCCGTTCTCCATT | 28 376<–28 358 | 52 | 16 975 | 16 938 | 37 | 16 938 | 100 | | |
| | US-CDC-N-1 | US | F | GACCCCAAAATCAGCGAAAT | 28 287–>28 306 | 40 | 16 987 | 16 970 | 17 | 16 970 | 100 | | [8,62] |
| | | | P | ACCCCGCGATTACGTTTGGTGGACC | 28 309–>28 332 | 72 | 16 955 | 16 647 | 308 | 16 920 | 98.4 | 1.6 | |
| | | | R | TCTGGTTACTGCCAGTTGAATCTG | 28 358<–28 335 | 48 | 16 979 | 16 876 | 103 | 16 876 | 100 | | |
| | US-CDC-N-2 | US | F | TTACAAACATTGGCCGCAAA | 29 164–>29 183 | 339 | 16 688 | 16 647 | 41 | 16 647 | 100 | | [8,62] |
| | | | P | ACAATTGCCCCCAGCGCTTCAG | 29 188–>29 210 | 351 | 16 676 | 16 605 | 71 | 16 605 | 100 | | |
| | | | R | GCGCGACATTCCGAAGAA | 29 230<–29 213 | 334 | 16 693 | 16 677 | 16 | 16 677 | 100 | | |
| | US-CDC-N-3 | US | F | GGGAGCCTTGAATACACCAAAA | 28 681–>28 702 | 63 | 16 964 | 16 747 | 217 | 16 943 | 98.8 | 1.2 | [8,62] |
| | | | P | AYCACATTGGCACCCGCAATCCTG | 28 704<–28 727 | 41 | 16 986 | 16 922 | 64 | 16 922 | 100 | | |
| | | | R | TGTAGCACGATTGCGAGCATTG | 28 752<–28 732 | 34 | 16 993 | 16 952 | 41 | 16 952 | 100 | | |
| | Young-N | Singapore | F | CTCAGTCCAAGATGGTATTTCT | 28 583–>28 604 | 67 | 16 960 | 16 953 | 7 | 16 953 | 100 | | [15] |
| | | | P | ACCTAGGGAACTGGCCCAGAAGCT | 28 608–>28 630 | 58 | 16 969 | 0 | 16 969 | 16 927 | 0.0 | 100 | |
| | | | R | AGCACCATAGGGAAGTCC | 28 648<–28 631 | 52 | 16 975 | 16 949 | 26 | 16 949 | 100 | | |
| | Corman-N | Germany | F | CAGATTGGCACCCGCAATC | 28 706–>28 724 | 38 | 16 989 | 16 954 | 35 | 16 954 | 100 | | [50] |
| | | | P | ACTTCCTCCAAGGAACAACATTGCCA | 28 754–>28 777 | 75 | 16 952 | 16 930 | 22 | 16 930 | 100 | | |
| | | | R | GAGGAACGAGAAGAGGCTTG | 28 833<–28 814 | 92 | 16 935 | 16 863 | 72 | 16 863 | 100 | | |
| | Won-N | South Korea | F | CAATGCTGCAATCGTGCTAC | 28 732–>28 751 | 33 | 16 994 | 16 953 | 41 | 16 953 | 100 | | [13] |
| | | | R | GTTGCGGACTACGTGATGAGG | 28 849<–28 830 | 85 | 16 942 | 16 788 | 154 | 16 788 | 100 | | |

(*Continued.*)

**Table 2.** (*Continued.*)

| gene target | assay name[a] | country | F/P/R[b] | sequence (5'–3') | position[c] | total number of sequences | | | | above threshold (≥0.5%)[d] | | | reference(s) |
|---|---|---|---|---|---|---|---|---|---|---|---|---|---|
| | | | | | | removed | informative | perfect match | with mismatches | total | perfect match (%) | with mismatches (%) | |
| | NIID-JP-N | Japan Japan | F | AAATTTGGGGACCAGGAAC | 29 125–>29 144 | 301 | 16 726 | 16 658 | 68 | 16 658 | 100 | | [8,51] |
| | | | P | ATGTCGCGGATTGGCATGGA | 29 222–>29 241 | 329 | 16 698 | 16 679 | 19 | 16 679 | 100 | | |
| | | | R | TGGCAGCTGTGTAGGTCAAC | 29 282<–29 263 | 309 | 16 718 | 0 | 16 718 | 16 687 | 0.0 | 100 | [51] |
| | | | R-v3 | TGGCACCTGTGTAGGTCAAC | 29 282<–29 263 | 309 | 16 718 | 16 687 | 31 | 16 687 | 100 | | |
| | HKU-N | Hong Kong | F | TAATCAGACAAGGAACTGATTA | 29 145–>29 166 | 309 | 16 718 | 16 667 | 51 | 16 667 | 100 | | [8,52] |
| | | | P | GCAAATTGTGGAATTTGCGG | 29 177<–29 196 | 347 | 16 680 | 16 637 | 43 | 16 637 | 100 | | |
| | | | R | CGAAGGTGTGACTTCCATG | 29 254<–29 236 | 320 | 16 707 | 16 668 | 39 | 16 668 | 100 | | |
| | Chan-N | China | F | GCGGTTCTTCGGAATGTCG | 29 210–>29 227 | 338 | 16 689 | 16 665 | 24 | 16 665 | 100 | | [9] |
| | | | P | AACGTGGTTGACCTACACAGST | 29 257–>29 278 | 311 | 16 716 | 16 680 | 36 | 16 680 | 100 | | |
| | | | R | TTGGATCTTTGTCATCCAATTTG | 29 306<–29 284 | 304 | 16 723 | 16 674 | 49 | 16 674 | 100 | | |

[a]The assays were named in the following format: organization/author-gene target.

[b] Forward primer (F), probe (P) and reverse primer (R).

[c] Positions shown are with reference to NC_045 512.2.

[d]A threshold of 0.5% was defined where only the sequence variants with greater than or equal to 0.5% incidence were further considered.

**Table 3.** Summary of primer/probe mismatches with SARS-CoV-2 genome.

| primer name | F/P/R[b] | sequence (5′–3′)[c] and suggested adjustment | genome position[d] | nucleotide primer | nucleotide genome | frequency |
|---|---|---|---|---|---|---|
| Charité-ORF1b | R | CARATGTTAAA**S**ACACTATTAGCATA<br>Suggested modification from S to A<br>(or R). CARATGTTAAA**A**ACACTATTAGCATA | 15 519 | S (G/C)[1] | T | 17 002/17 002 (100%) |
| Chan-ORF1ab | P | TTAAGATGTGGTG**C**TTGCATACGTAGAC | 16 289 | C | T | 144/16 930 (0.9%) |
|  | R | **G**TGTGATGTTGAWATGACATGGTC<br>Suggested modification from G to A<br>**A**TGTGATGTTGAWATGACATGGTC | 16 353 | C[a] | T | 16 932/16 932 (100%) |
| CN-CDC-N | F | **GGG**GAACTTCTCCTGCTAGAAT | 28 881<br>28 882<br>28 883 | GGG | AAC | 3129/16 662 (18.8%) |
| US-CDC-N-1 | P | AC**C**CCGCCATTACGTTTGGTGGACC | 29 311 | C | T | 273/16 920 (1.6%) |
| US-CDC-N-3 | F | GGGAGCC**T**TGAATACACCAAAA | 28 688 | T | C | 196/16 943 (1.2%) |
| Young-N | P | ACCTAGGAACTGG**CC**CAGAAGCT<br>Suggested modification from C to G<br>ACCTAGGAACTGG**GC**CAGAAGCT | 28 621 | C | G | 16 969/16 969 (100%) |
| NIID-JP-N | R | TGGCA**G**TGTGTAGGTCAAC<br>Suggested modification from G to C [51]<br>TGGCA**C**TGTGTAGGTCAAC | 29 277 | C[a] | G | 16 687/16 687 (100%) |

[a]Reverse-complemented.

[b]Forward primer (F), probe (P) and reverse primer (R).

[c]Underlined and bold sequences indicate the mismatch observed and the suggested adjustment.

[d]Positions shown are with reference to NC_045512.2.

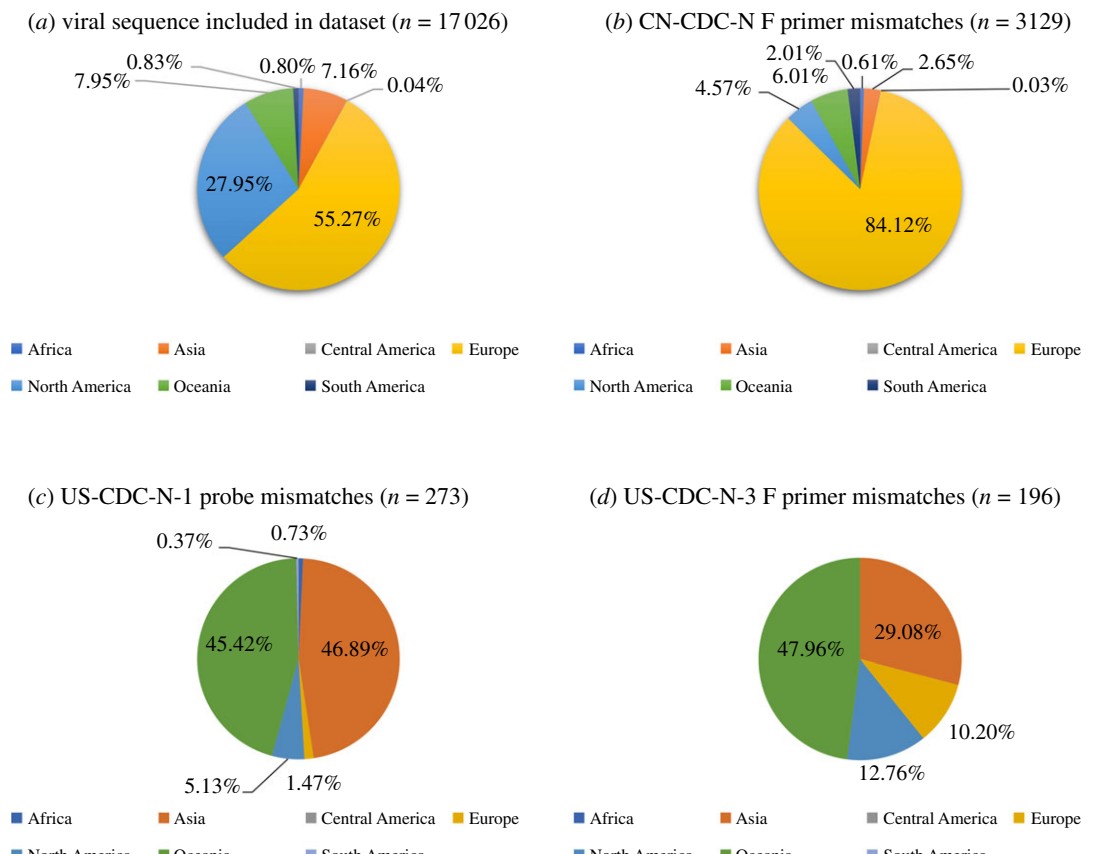

**Figure 4.** Geographical distribution of included sequences dataset (*a*) and mismatches for CN-CDC-N forward primer (*b*), US-CDC-N-1 probe (*c*) and US-CDC-N-3 forward primer (*d*). The total number of sequences in each dataset is given in parentheses. Data used to draw graphs are included in electronic supplementary material, file S3.

correcting the mismatches between primers/probes and template may help to improve the sensitivity of certain diagnostic assays.

There have been recent efforts along the same line where a limited number of viral sequences were aligned with primers/probes to search for mismatches. One of the recent preprints used 992 sequences to report some variants in the primer/probe binding regions [72]. However, many of the mismatches could be rare variants or sequencing errors, and variability in the assay binding regions should be assessed across a larger number of viral sequences. In addition, the diagnostic assay should not be revised based on the presence of rare variants in the population and thus a threshold of 0.5% was defined to eliminate such variants from the analysis. Some of the mismatches observed by this preprint were confirmed in the larger dataset of the current study. Other variants were not observed or did not reach the threshold and thus were not reported in the final analysis. It cannot be excluded that empirical threshold adjustment of this study might have missed some significant variants. For instance, choosing a threshold of 0.2% would have resulted in a mismatch with five additional assays that were reported to match with 100% of sequences in the current analysis. Another recent preprint reported a bioinformatics system named 'BioLaboro' to assess the efficacy of the existing PCR assays to detect pathogens as they evolve [75]. However, this system requires specialized software and large RAM hardware which is not generally available in regular diagnostic or research laboratories. By contrast, the current study validates a pipeline for *in silico* re-evaluation of PCR diagnostic assays of SARS-CoV-2. This approach has successfully been applied previously for influenza A virus [61]. Using freely available open-source software, the analysis was performed on a regular desktop computer without any need for special hardware. The pipeline does not require extensive computational skills except for some sequence alignment skills. The pipeline can be applied to a SARS-CoV-2 diagnostic assay of choice.

Verification of *in silico* nucleotide identity match, termed as inclusivity analysis, is also a component of the performance criteria of COVID-19 diagnostic assays by the U.S. Food and Drug Administration (FDA) as well as the European Commission [76,77]. Several commercially developed COVID-19 diagnostic assays have received limited regulatory approval due to the emergency situation. As of 12

May 2020, a total of 54 commercial diagnostic test kits including the one developed by the US-CDC have received emergency use authorization (EUA) from the FDA [78]. The CDC has also reported one nucleotide mismatch in the N1 forward primer in their inclusivity assay using sequences available as of 1 February 2020 [62]. Some commercial kits like BD BioGX use CDC primers and thus do not conduct independent inclusivity analysis [79]. Many other kits have reported the alignment of their assay primers/probes with a couple of hundred sequences [80–85]. As primer/probe identity for most commercial kits is not revealed, manufacturer-independent data are scarce. Recent comparisons of SARS-CoV-2 diagnostic assays have shown some discordance which may partially be due to sequence differences [86,87]. Therefore, there is a need for comprehensive inclusivity assessment of commercial diagnostic assays. Although not addressed in this article, other factors for reassessment include *in silico* cross-reactivity with human genes, genes of other members of family Coronaviridae and other respiratory viruses/bacteria.

The methodology outlined here uses MSA of publicly available viral sequences and is prone to certain biases despite its general utility in diagnostic PCR assay design. One of the biases is the compositional bias, which may arise as a result of sampling from certain geographical locations due to access to better facilities for viral genome sequencing or location of the outbreak. Based on a relatively moderate mutation rate in the genome, the results obtained can be applied globally, but caution should be exercised when drawing conclusions from the results for a specific region, especially with a smaller number of sequences included. Another possible geographical bias can arise due to the removal of data collected from certain countries or regions. However, the fact that less than 2.1% of sequences were removed for 73 out of 76 primers/probes studied mitigates this concern in the current study. The geographical analysis of the removed data (approx. 6%) of the remaining three primers/probes showed that most of the removed viral sequences were from Europe as expected (electronic supplementary material, file S4). Although the risk of data skew geographically cannot be ruled out completely, this much data exclusion is in line with previous reports [61]. Another source of compositional bias may be the redundancy where the same viral strain is re-sequenced and re-submitted to the sequence database.

Another source of bias may arise from the submission of isolates after passaging in the cell culture as well as sequencing artefacts including ambiguous data, short artificial insertions or deletions, incorrect sequence directions, incorrect nucleotide insertions, short sequence stretches and sequence longer than standard length [88]. Most data in the EpiCov database include the full-length data, and thus short sequences were not included in the study. To remove artificially inserted sequences and sequences at the ends, if any, MSA was performed with the option to keep the alignment length according to the reference sequence. In this methodology, no gaps are inserted in the reference sequence and corresponding sites in the other sequences are deleted. Therefore, this methodology can potentially remove any real insertions as well. However, only seven insertions affecting 31 sequences are catalogued in CoV-GLUE database (http://cov-glue.cvr.gla.ac.uk/#/insertion) as of 22 May 2020 [89]. The use of SequenceTracer in the tracing pipeline successfully filters out ambiguous data and deletions [61]. As SequenceTracer removes all the sequences with short and missing sequences, a real deletion of a stretch of sequence would also be filtered out. However, only a few sequences were removed in the 'outgroup2' or in 'excluded' group (figures 2 and 3; electronic supplementary material, file S2). In line, none of the deletions affecting more than two sequences listed in CoV-GLUE database (http://cov-glue.cvr.gla.ac.uk/#/deletion) as of 22 May 2020 were found in the ROI under study.

# 5. Conclusion

This work outlines a comprehensive approach for the bioinformatics reassessment of PCR diagnostic assays for SARS-CoV-2. The application of this strategy on 27 previously developed assays using 17 027 viral sequences showed mutations/mismatches in primer/probe binding regions of seven assays. This information will act as a reference and may help re-evaluate COVID-19 diagnostic strategies. *In silico* analysis of primers/probes should be coupled with empirical testing on clinical samples and the primers/probes that work well *in silico* as well as empirically should be used in a diagnostic assay for SARS-CoV-2.

Data accessibility. A list of accession numbers of sequences is included in electronic supplementary material, file S1. Sequence tracing figures of all the assays not shown in the main article are included in electronic supplementary material, file S2. Geographical data used to draw graphs in figure 4 are included in electronic supplementary

material, file S3. The geographical analysis of removed data for three primers/probe with the highest frequency is included in electronic supplementary material, file S4.

Authors' contributions. K.A.K. conceived and designed the study, carried out sequence alignments, performed data analysis and drafted the manuscript. P.C. provided valuable suggestions throughout, critically revised the manuscript and arranged the funding for the project.

Competing interests. The authors have no competing interests.

Funding. Funding for this study was provided by the Canadian Institutes of Health Research operating (grant no. RN227427–324983) awarded to P.C.

Acknowledgements. We gratefully acknowledge the great work of authors, originating and submitting laboratories of the sequences from GISAID's EpiCoV™ Database on which this research is based. The list is included in electronic supplementary material, file S1. We thank Alexander Nagy (State Veterinary Institute, Prague, Czech Republic) for critical reading of the manuscript.

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
