## [Reviewer comments · Royal Society Open Science]

Review History

RSOS-200636.R0 (Original submission)

Review form: Reviewer 1

Do you have any ethical concerns with this paper?

No

Recommendation?

Accept with minor revision

Comments to the Author(s)

The research question is scientifically valid. As the SARS-CoV-2 virus evolves over the course of the pandemic, there is potential for mutations to arise in the primer/probe binding regions, which could lead to false negative diagnostic test results. The author proposes to assess for genetic variability in these diagnostic target sequences, and thereby the current potential for false negative results, based on a large publicly available database of viral genome sequences that have been submitted from global sources.

The hypothesis that there may be existing differences in SARS-CoV-2 primer/probe binding sites among circulating viruses is plausible. The rationale for testing this hypothesis is well explained:

1. There are already demonstrated issues of false negative PCR test results for COVID-19; 2.

Issues with primer/probe mismatch have led to false negative diagnostic results for several other viruses (for which the author provides many relevant references); 3. The regular verification of the primer/probe binding sites for COVID-19 diagnosis has been recommended by the American Society for Microbiology.

The proposed analysis pipeline seems appropriate and feasible. This study makes use of a valuable public resource, i.e. the large public EpiCoV database of virus genome sequences available through the GISAID. The choice of diagnostic primer/probe sets to test seems to encompass those that are currently most relevant and being implemented on a large scale. The author plans to make accession numbers of included genomes available, which will be important supplemental material to enable others to replicate study. While not specifically mentioned, providing the program versions and specific commands used would also be valuable.

The following are some minor suggestions for improved clarity:

1. The abstract should specify that the >9000 viral genome sequences are publicly available.
2. The author could more clearly state the anticipated relevance in both the abstract and introduction. If the results are negative (i.e. there are no mutations found in currently used primer/probe PCR target sequences among globally circulating strains), this provides a higher degree of confidence in the test results, and information that these diagnostic targets are highly conserved. If, alternatively, differences are found, this would a) help identify geographic regions where false negative diagnostic results may have been occurring; b) help guide the preferential use of certain diagnostic targets over others, and c) provide further information on the mutation rate in different parts of the virus genome.
3. The introduction refers to protocols that were published by the WHO, however the placement of reference 8 is misleading, as this is a review paper and not a WHO publication. This should be rephrased to clearly identify what work the reference alludes to.
4. The author could explain more clearly why the presence of a proofreading exonuclease is relevant (i.e. it would theoretically reduce errors during replication/ enhance CoV replication fidelity).
5. The objective could be rephrased to "... re-assessment of the currently recommended PCR primers/probes for COVID-19 diagnostics through the evaluation of ..."
6. The manuscript would benefit from English language edits, particularly grammar. Organisation names should be checked (e.g. US Centers for Disease Control is incorrectly referred to as CDC USA). There is also inconsistent and sometimes inappropriate use of upper versus lower case for virus names.
7. Some references are missing information [23] and/or require reformatting [29].

Review form: Reviewer 2

Do you have any ethical concerns with this paper?

No

Recommendation?

Reject

Comments to the Author(s)

Lacks novelty. It is not very clear how this work adds to advancement of understanding of COVID-19

Review form: Reviewer 3

Do you have any ethical concerns with this paper?

No

Recommendation?

Accept with minor revision

Comments to the Author(s)

To evaluate the diagnostic primers for SARS-CoV-2 is an important and timely topic. And Dr. Khan has proposed a logical and efficient way to do so. The method is feasible, with the majority of them will be from open-sources.

However, the proposal and methodology are still quite general. It could be further improved by clearer addressing the sub-questions intended to answer. For example, just some of my thoughts, one of the questions could be how the primers available now have mismatches with other viruses, such as influenza, etc. Another one could be investigating primer efficiency cross different SARS-CoV-2 lines as several mutations have been reported recently even within the infectious domains. Finally, it would also be interesting to line the results with the diagnostic situation among different countries.

Overall, the author proposed a much-needed evaluation in the field, and the methods proposed are reasonably quick to do. But, before starting, the proposal would benefit from more detailed planning of what questions to answer.

Review form: Reviewer 4

Do you have any ethical concerns with this paper?

No

Recommendation?

Accept with minor revision

Comments to the Author(s)

The author proposes to identify mutations in SARS-CoV-2 genomes that overlap with binding sites for primers and probes used in published RT-PCR assays. To do this, the author would download available SARS-CoV-2 genomes and perform multiple sequence alignment (MSA) with these genomes and primer/probe sets to identify variable regions. Web-based tools would be used to filter and analyze the results of the MSA. This is a valid study, and while the results of this analysis would be of interest, another study has already demonstrated that mutations exist in SARS-CoV-2 at primer/probe binding regions (<https://www.medrxiv.org/content/10.1101/2020.03.30.20048108v1.full.pdf>). This paper would need to be addressed in the discussion.

Specific responses:

In regard to scientific validity, the research question in Table 1 about the existence of mutations in PCR primers/probes for SARS-CoV-2 has already been answered in the above paper.

Nonetheless, that paper only scanned 992 genomes, and there would likely be benefit to doing a larger scale scan as proposed by this paper.

The hypothesis stated is that mutations can cause mismatches in the primer/probe region. This hypothesis is logical but again has already been investigated in the above study.

The analyses proposed are methodologically sound and feasible. They follow a study published recently.

There could be additional clarity around the specific parameters to be used in the multiple sequence alignment. Specifying these parameters would be important for reproducibility.

The author has included a positive control that would enable proper testing of the hypothesis.

Decision letter (RSOS-200636.R0)

Dear Dr Khan,

The Editors assigned to your Stage 1 Registered Report ("Sequence variation in PCR primers/probe binding regions in coronavirus SARS-CoV-2 genome") have now received comments from reviewers. We would like you to revise your paper in accordance with the referee and editors suggestions which can be found below (not including confidential reports to the Editor). Please note this decision does not guarantee eventual acceptance.

When submitting your revised manuscript, you must respond to the comments made by the referees and upload a file "Response to Referees" in "Section 2 - File Upload". Please use this to document how you have responded to the comments, and the adjustments you have made. In order to expedite the processing of the revised manuscript, please be as specific as possible in your response.

Kind regards,
Andrew Dunn
Royal Society Open Science
openscience@royalsociety.org

on behalf of Professor Chris Chambers (Registered Reports Editor, Royal Society Open Science)
openscience@royalsociety.org

Associate Editor Comments to Author (Professor Chris Chambers):

Four reviewers and a specialist in-house editor have now assessed the manuscript. Overall the reviewers find merit in the proposal, while also noting areas in need of attention, including the clarity of the rationale, level of methodological detail, and potential risk of bias in the analysis. In particular, Reviewers 1 and 3 are overall positive but note the need for clarification of the rationale and specific questions being addressed in the proposal. Meanwhile, Reviewer 4 points to relevant existing literature that should be considered and also notes a lack of methodological detail concerning the parameters for multiple sequence alignment. On this latter point especially,

please ensure that all study procedures and analyses are explained with sufficient detail to be fully reproducible.

In addition, the specialist editor notes that by limiting the sampling to high quality complete sequences the you could end up with a geographically biased sample if the samples are of systematically higher quality in some countries than others. So while both the quality criterion and the geographic sampling are important, they could potentially conflict. Please address this issue carefully in the revision.

Each of the reviewers also raise a number of specific points (including need for careful proofreading for grammar), so please ensure that the revision is comprehensive and addresses every point raised. Finally, I note that you need not respond to Reviewer 2. I include the review in the interests of completeness but it contains no information that is relevant to the assessment of a Stage 1 Registered Report.

Comments to Author:

Reviewer: 1

Comments to the Author(s)

The research question is scientifically valid. As the SARS-CoV-2 virus evolves over the course of the pandemic, there is potential for mutations to arise in the primer/probe binding regions, which could lead to false negative diagnostic test results. The author proposes to assess for genetic variability in these diagnostic target sequences, and thereby the current potential for false negative results, based on a large publicly available database of viral genome sequences that have been submitted from global sources.

The hypothesis that there may be existing differences in SARS-CoV-2 primer/probe binding sites among circulating viruses is plausible. The rationale for testing this hypothesis is well explained: 1. There are already demonstrated issues of false negative PCR test results for COVID-19; 2. Issues with primer/probe mismatch have led to false negative diagnostic results for several other viruses (for which the author provides many relevant references); 3. The regular verification of the primer/probe binding sites for COVID-19 diagnosis has been recommended by the American Society for Microbiology.

The proposed analysis pipeline seems appropriate and feasible. This study makes use of a valuable public resource, i.e. the large public EpiCoV database of virus genome sequences available through the GISAID. The choice of diagnostic primer/probe sets to test seems to encompass those that are currently most relevant and being implemented on a large scale. The author plans to make accession numbers of included genomes available, which will be important supplemental material to enable others to replicate study. While not specifically mentioned, providing the program versions and specific commands used would also be valuable.

The following are some minor suggestions for improved clarity:

1. The abstract should specify that the >9000 viral genome sequences are publicly available.
2. The author could more clearly state the anticipated relevance in both the abstract and introduction. If the results are negative (i.e. there are no mutations found in currently used primer/probe PCR target sequences among globally circulating strains), this provides a higher degree of confidence in the test results, and information that these diagnostic targets are highly conserved. If, alternatively, differences are found, this would a) help identify geographic regions where false negative diagnostic results may have been occurring; b) help guide the preferential use of certain diagnostic targets over others, and c) provide further information on the mutation rate in different parts of the virus genome.
3. The introduction refers to protocols that were published by the WHO, however the placement of reference 8 is misleading, as this is a review paper and not a WHO publication. This should be rephrased to clearly identify what work the reference alludes to.
4. The author could explain more clearly why the presence of a proofreading exonuclease is relevant (i.e. it would theoretically reduce errors during replication/ enhance CoV replication fidelity).

5. The objective could be rephrased to "... re-assessment of the currently recommended PCR primers/probes for COVID-19 diagnostics through the evaluation of ..."
6. The manuscript would benefit from English language edits, particularly grammar. Organisation names should be checked (e.g. US Centers for Disease Control is incorrectly referred to as CDC USA). There is also inconsistent and sometimes inappropriate use of upper versus lower case for virus names.
7. Some references are missing information [23] and/or require reformatting [29].

Reviewer: 2

Comments to the Author(s)

Lacks novelty. It is not very clear how this work adds to advancement of understanding of COVID-19

Reviewer: 3

Comments to the Author(s)

To evaluate the diagnostic primers for SARS-CoV-2 is an important and timely topic. And Dr. Khan has proposed a logical and efficient way to do so. The method is feasible, with the majority of them will be from open-sources.

However, the proposal and methodology are still quite general. It could be further improved by clearer addressing the sub-questions intended to answer. For example, just some of my thoughts, one of the questions could be how the primers available now have mismatches with other viruses, such as influenza, etc. Another one could be investigating primer efficiency cross different SARS-CoV-2 lines as several mutations have been reported recently even within the infectious domains. Finally, it would also be interesting to line the results with the diagnostic situation among different countries.

Overall, the author proposed a much-needed evaluation in the field, and the methods proposed are reasonably quick to do. But, before starting, the proposal would benefit from more detailed planning of what questions to answer.

Reviewer: 4

Comments to the Author(s)

The author proposes to identify mutations in SARS-CoV-2 genomes that overlap with binding sites for primers and probes used in published RT-PCR assays. To do this, the author would download available SARS-CoV-2 genomes and perform multiple sequence alignment (MSA) with these genomes and primer/probe sets to identify variable regions. Web-based tools would be used to filter and analyze the results of the MSA. This is a valid study, and while the results of this analysis would be of interest, another study has already demonstrated that mutations exist in SARS-CoV-2 at primer/probe binding regions (<https://www.medrxiv.org/content/10.1101/2020.03.30.20048108v1.full.pdf>). This paper would need to be addressed in the discussion.

Specific responses:

In regard to scientific validity, the research question in Table 1 about the existence of mutations in PCR primers/probes for SARS-CoV-2 has already been answered in the above paper. Nonetheless, that paper only scanned 992 genomes, and there would likely be benefit to doing a larger scale scan as proposed by this paper.

The hypothesis stated is that mutations can cause mismatches in the primer/probe region. This hypothesis is logical but again has already been investigated in the above study.

The analyses proposed are methodologically sound and feasible. They follow a study published recently.

There could be additional clarity around the specific parameters to be used in the multiple sequence alignment. Specifying these parameters would be important for reproducibility.

The author has included a positive control that would enable proper testing of the hypothesis.

Specialist editor comments:

Comments to the Author(s)

I think the rationale for this research is clear and I agree that the information is valuable, no matter what outcome will be presented.

My major concern is that while the authors specify that they plan to use sequences from a number of countries, I didn't see a proper interpretation of the information in the text. Let me explain what I mean: On one hand, the authors plan to use sequences originating from patients in different countries (which I think is good). On the other hand, they plan to use only high quality complete sequences (which again is good). I am not sure that these two statements necessarily overlap, and I am a little worried that this analysis could be biased based on that. I think that the authors should clearly address the geographical distribution and the potential biases.

Author's Response to Decision Letter for (RSOS-200636.R0)

See Appendix A.

Decision letter (RSOS-200636.R1)

Dear Dr Khan

On behalf of the Editor, I am pleased to inform you that your Manuscript RSOS-200636.R1 entitled "Sequence variation in PCR primers/probe binding regions of coronavirus SARS-CoV-2 genome" has been accepted in principle for publication in Royal Society Open Science. Your revised manuscript was assessed by a specialist editor on the Biochemistry, Cellular and Molecular Biology editorial team and by the Registered Reports editor. Together we concluded that you had addressed the reviewers' comments sufficiently to achieve Stage 1 in-principle acceptance.

Please read the following email carefully

Your accepted Stage 1 manuscript has been publicly registered at:
<https://doi.org/10.17605/OSF.IO/YM8GC>

Following completion of your study, we invite you to resubmit your paper for peer review as a Stage 2 Registered Report. Please note that your manuscript can still be rejected for publication at Stage 2 if the Editors consider any of the following conditions to be met:

- The results were unable to test the authors' proposed hypotheses by failing to meet the approved outcome-neutral criteria.
- The authors altered the Introduction, rationale, or hypotheses, as approved in the Stage 1 submission.
- The authors failed to adhere closely to the registered experimental procedures. Please note that any deviations from the approved experimental procedures must be communicated to the editor immediately for approval, and prior to the completion of data collection. Failure to do so can result in revocation of in-principle acceptance and rejection at Stage 2 (see complete guidelines for further information).
- Any post-hoc (unregistered) analyses were either unjustified, insufficiently caveated, or overly dominant in shaping the authors' conclusions.
- The authors' conclusions were not justified given the data obtained.

We encourage you to read the complete guidelines for authors concerning Stage 2 submissions at <https://royalsocietypublishing.org/rsos/registered-reports#ReviewerGuideRegRep>. Please especially note the requirements for data sharing, reporting the URL of the independently registered protocol, and that withdrawing your manuscript will result in publication of a Withdrawn Registration.

Once again, thank you for submitting your manuscript to Royal Society Open Science and we look forward to receiving your Stage 2 submission. If you have any questions at all, please do not hesitate to get in touch. We look forward to hearing from you shortly with the anticipated submission date for your stage two manuscript.

on behalf of Professor Chris Chambers (Registered Reports Editor, Royal Society Open Science)
openscience@royalsociety.org

Author's Response to Decision Letter for (RSOS-200636.R1)

See Appendix B.

RSOS-200636.R2 (Revision)

Review form: Reviewer 1

Is the manuscript scientifically sound in its present form?

Yes

Are the interpretations and conclusions justified by the results?

Yes

Is the language acceptable?

No

Do you have any ethical concerns with this paper?

No

Have you any concerns about statistical analyses in this paper?

No

Recommendation?

Accept with minor revision

Comments to the Author(s)

The Introduction, rationale and stated hypotheses are the same as those approved at Stage 1 submission. However, the title has changed. The term "Inclusivity analysis" is a bit confusing, both in the title and throughout the paper. It should be properly defined in the text, and ideally a reference provided. I assume that it means something along the lines of 'proportion of the sequenced virus isolates that would be expected to be detected by a given assay, based on perfect matches in the primer/probe binding sequences'?

The experimental procedures were carried out as planned. The only exception was that base composition of each nucleotide position was not analyzed, although this was justified based on their findings of moderate sequence variation.

Additional methods details are provided in comparison to the original proposal. These are presented as part of the "Results" section, where they more clearly outline some of the steps taken. These are still in line with the original proposal. The one potential exception is the introduction of a threshold for inclusivity analysis. This concept could be explained in slightly more detail. I assume the authors mean that a variant was only considered if it was present in > 0.5% of the informative aligned sequences (i.e. > ~85 isolates, depending on the MSA)? This seems quite lenient, i.e. you might miss important variants, particularly if the same mutations are found from isolates sequenced independently in different labs. However, I think it is fine to keep this threshold, but providing a bit more discussion on this subject, and defining the threshold more clearly. This concept is generally clear from the Tables and Figures, but less so from the text. The author's conclusions are justified given the data.

Suggested minor edits:

In general, the manuscript would benefit from English language edits, particularly the most recently added sections. Data should be referred to as plural throughout. In many cases, this is also true for 'assay'.

Summary:

-Line 27: add comma after assays, and change 'the ones' to 'those'

-Lines 28-29: add long dashes before 'performed' and after 'world' to help break up this long sentence

-Line 34: change 'decision-makers' to 'policy-makers'

Methods:

- p3 Line 15: unsure of the wording “were stratified to using”... should it be “stratified using...”?
- p3 Lines 21-25: This sentence needs rewording.

Results

- 4.1 Lines 32-39: This is more appropriate material for the Introduction, and does not represent results of this study. Suggest either moving to Introduction or removing completely.
- 4.1 Lines 41-47: This would be more appropriate in the Methods section.
- 4.2 p3 Lines 49- p4 Line 12: This would also be more appropriate in Methods section.
- p3 Line 18 – It’s not quite sure what is meant by “one mismatch with all the sequences above threshold”. Perhaps the terms “above threshold” and “informative sequences” are being mixed?

Discussion:

- p5 Line 11 – add the word “data” after “genomic sequence” (i.e. limited genomic sequence data)
- p5 Line 19 – change “sensitivity of the diagnostic assays” to “sensitivity of certain diagnostic assays”.
- p5 Line 21 – change to “along the same line” (remove ‘with’)
- p5 Line 25 – change “accessed” to “assessed”
- p5 Line 28 – change “reprint” to “preprint”
- p6 Lines 1-2 – it would be helpful to specify that most of the sequences originating in this study were specifically from UK and US (rather than just Europe and North America).
- p6 Lines 2-3 – remove the word “were” before ‘originated’ and before ‘from Africa’.

Figures:

- Figure 1 caption – the symbols explaining the specific parts of the Figure are switched.
- Figure 4a title – should be ‘sequences’ (plural)
- For each panel in Figure 4, it would be helpful to provide the total number of isolates (denominator)

Tables:

- Table 1 caption – you should specify what the threshold is

References

- Some references are missing information [5, 12, 14, etc.] and should be checked for proper formatting.

Supp File 2

- should specify the reference sequence used so that the document is more ‘stand-alone’

Review form: Reviewer 3

Is the manuscript scientifically sound in its present form?

Yes

Are the interpretations and conclusions justified by the results?

Yes

Is the language acceptable?

Yes

Do you have any ethical concerns with this paper?

No

Have you any concerns about statistical analyses in this paper?

No

Recommendation?

Accept with minor revision

Comments to the Author(s)

The paper is well-organised and I am glad to see that the author has addressed my concerns. The introduction, rationale and hypotheses are the same as stage1. The results and conclusions are sound.

However, the result parts were a bit wonky and detailed descriptions of analysed data were missing, such as the pie charts provided on page 20.

Also, the reason under the chosen threshold was firstly described in the discussion part, which needs to be carefully explained before in the result part, along with how does threshold change affect the whole outcome need to be explained.

Decision letter (RSOS-200636.R2)

Dear Dr Khan:

On behalf of the Editor, I am pleased to inform you that your Stage 2 Registered Report RSOS-200636.R2 entitled "Inclusivity analysis of diagnostic PCR assays through evaluation of sequence variation in coronavirus SARS-CoV-2 genome" has been deemed suitable for publication in Royal Society Open Science subject to revision in accordance with the referee suggestions. Please find the referees' comments at the end of this email as well as annotated PDF from the Registered Reports subject editor (Chris Chambers).

The reviewers and Subject Editor have recommended publication, but also suggest some revisions to your manuscript. Therefore, I invite you to respond to the comments and revise your manuscript.

Please also ensure that all the below editorial sections are included where appropriate -- if any section is not applicable to your manuscript, please can we ask you to nevertheless include the heading, but explicitly state that the heading is inapplicable. An example of these sections is attached with this email.

- Ethics statement

- Data accessibility

It is a condition of publication that all supporting data are made available either as supplementary information or preferably in a suitable permanent repository. The data accessibility section should state where the article's supporting data can be accessed. This section

should also include details, where possible of where to access other relevant research materials such as statistical tools, protocols, software etc can be accessed. If the data has been deposited in an external repository this section should list the database, accession number and link to the DOI for all data from the article that has been made publicly available. Data sets that have been deposited in an external repository and have a DOI should also be appropriately cited in the manuscript and included in the reference list.

If you wish to submit your supporting data or code to Dryad (<http://datadryad.org/>), or modify your current submission to dryad, please use the following link:
[http://datadryad.org/submit?journalID=RSOS&manu=\(Document not available\)](http://datadryad.org/submit?journalID=RSOS&manu=(Document not available))

- **Competing interests**

- **Authors' contributions**

- **Acknowledgements**

- **Funding statement**

1) A text file of the manuscript (tex, txt, rtf, docx or doc), references, tables (including captions) and figure captions. Do not upload a PDF as your "Main Document".

- 2) A separate electronic file of each figure (EPS or print-quality PDF preferred (either format should be produced directly from original creation package), or original software format)
- 3) Included a 100 word media summary of your paper when requested at submission. Please ensure you have entered correct contact details (email, institution and telephone) in your user account
- 4) Included the raw data to support the claims made in your paper. You can either include your data as electronic supplementary material or upload to a repository and include the relevant doi within your manuscript
- 5) All supplementary materials accompanying an accepted article will be treated as in their final form. Note that the Royal Society will neither edit nor typeset supplementary material and it will be hosted as provided. Please ensure that the supplementary material includes the paper details where possible (authors, article title, journal name).

on behalf of Professor Chris Chambers
(Registered Reports Editor, Royal Society Open Science)
openscience@royalsociety.org

Associate Editor Comments to Author (Professor Chris Chambers):

Associate Editor: 1

Comments to the Author:

Two of the original Stage 1 reviewers have now reappraised the Stage 2 manuscript, in addition to a specialist editor on the Biochemistry, Cellular and Molecular Biology editorial board, and the Registered Reports editor.

Both reviewers are broadly satisfied with the Stage 2 submission while also noting several areas requiring clarification. Reviewer 1 notes the need for a clearer definition and justification of the term "inclusivity analysis" (especially given that the term is now in the title), and both reviewers question the rationale for the 0.5% threshold, which appears to have been decided after Stage 1 acceptance. The selection of this threshold -- and the point in time that it was decided (before or after observation of the data) -- needs to be made clear.

In addition to the reviews, I have passed through the manuscript and made detailed suggestions for edits to improve grammar, clarity and presentation. Please find attached an annotated PDF with these additional comments and suggestions.

Comments to Author:

Reviewer: 1

Comments to the Author(s)

The Introduction, rationale and stated hypotheses are the same as those approved at Stage 1 submission. However, the title has changed. The term “Inclusivity analysis” is a bit confusing, both in the title and throughout the paper. It should be properly defined in the text, and ideally a reference provided. I assume that it means something along the lines of ‘proportion of the sequenced virus isolates that would be expected to be detected by a given assay, based on perfect matches in the primer/probe binding sequences’?

The experimental procedures were carried out as planned. The only exception was that base composition of each nucleotide position was not analyzed, although this was justified based on their findings of moderate sequence variation.

Additional methods details are provided in comparison to the original proposal. These are presented as part of the “Results” section, where they more clearly outline some of the steps taken. These are still in line with the original proposal. The one potential exception is the introduction of a threshold for inclusivity analysis. This concept could be explained in slightly more detail. I assume the authors mean that a variant was only considered if it was present in > 0.5% of the informative aligned sequences (i.e. > ~85 isolates, depending on the MSA)? This seems quite lenient, i.e. you might miss important variants, particularly if the same mutations are found from isolates sequenced independently in different labs. However, I think it is fine to keep this threshold, but providing a bit more discussion on this subject, and defining the threshold more clearly. This concept is generally clear from the Tables and Figures, but less so from the text. The author's conclusions are justified given the data.

Suggested minor edits:

In general, the manuscript would benefit from English language edits, particularly the most recently added sections. Data should be referred to as plural throughout. In many cases, this is also true for ‘assay’.

Summary:

- Line 27: add comma after assays, and change ‘the ones’ to ‘those’
- Lines 28-29: add long dashes before ‘performed’ and after ‘world’ to help break up this long sentence
- Line 34: change ‘decision-makers’ to ‘policy-makers’

Methods:

- p3 Line 15: unsure of the wording “were stratified to using”... should it be “stratified using...”?
- p3 Lines 21-25: This sentence needs rewording.

Results

- 4.1 Lines 32-39: This is more appropriate material for the Introduction, and does not represent results of this study. Suggest either moving to Introduction or removing completely.
- 4.1 Lines 41-47: This would be more appropriate in the Methods section.
- 4.2 p3 Lines 49- p4 Line 12: This would also be more appropriate in Methods section.
- p3 Line 18 – It’s not quite sure what is meant by “one mismatch with all the sequences above threshold”. Perhaps the terms “above threshold” and “informative sequences” are being mixed?

Discussion:

- p5 Line 11 – add the word “data” after “genomic sequence” (i.e. limited genomic sequence data)
- p5 Line 19 – change “sensitivity of the diagnostic assays” to “sensitivity of certain diagnostic assays”.
- p5 Line 21 – change to “along the same line” (remove ‘with’)
- p5 Line 25 – change “accessed” to “assessed”
- p5 Line 28 – change “reprint” to “preprint”

- p6 Lines 1-2 – it would be helpful to specify that most of the sequences originating in this study were specifically from UK and US (rather than just Europe and North America).
- p6 Lines 2-3 – remove the word “were” before ‘originated’ and before ‘from Africa’.

Figures:

- Figure 1 caption – the symbols explaining the specific parts of the Figure are switched.
- Figure 4a title – should be ‘sequences’ (plural)
- For each panel in Figure 4, it would be helpful to provide the total number of isolates (denominator)

Tables:

Table 1 caption – you should specify what the threshold is

References

-Some references are missing information [5, 12, 14, etc.] and should be checked for proper formatting.

Supp File 2

-should specify the reference sequence used so that the document is more ‘stand-alone’

Reviewer: 3

Comments to the Author(s)

The paper is well-organised and I am glad to see that the author has addressed my concerns. The introduction, rationale and hypotheses are the same as stage1. The results and conclusions are sound.

However, the result parts were a bit wonky and detailed descriptions of analysed data were missing, such as the pie charts provided on page 20.

Also, the reason under the chosen threshold was firstly described in the discussion part, which needs to be carefully explained before in the result part, along with how does threshold change affect the whole outcome need to be explained.

Author's Response to Decision Letter for (RSOS-200636.R2)

See Appendix C.

Decision letter (RSOS-200636.R3)

Dear Dr Khan:

It is a pleasure to accept your manuscript entitled "Presence of mismatches between diagnostic PCR assays and coronavirus SARS-CoV-2 genome" in its current form for publication in Royal Society Open Science.

Royal Society Open Science Editorial Office
openscience@royalsociety.org

on behalf of Professor Chris Chambers (Subject Editor)
openscience@royalsociety.org

Appendix A

Response to Referees:

Reviewer: 1

While not specifically mentioned, providing the program versions and specific commands used would also be valuable.

Response: The Program versions have been updated in Methods section. In addition, specific commands used in different programs have been added in the pipeline summary (Fig 1).

The following are some minor suggestions for improved clarity:

1. The abstract should specify that the >9000 viral genome sequences are publicly available.

Response: Modified as suggested

2. The author could more clearly state the anticipated relevance in both the abstract and introduction. If the results are negative (i.e. there are no mutations found in currently used primer/probe PCR target sequences among globally circulating strains), this provides a higher degree of confidence in the test results, and information that these diagnostic targets are highly conserved. If, alternatively, differences are found, this would a) help identify geographic regions where false negative diagnostic results may have been occurring; b) help guide the preferential use of certain diagnostic targets over others, and c) provide further information on the mutation rate in different parts of the virus genome.

Response: The abstract and the Introduction has been modified with anticipated relevance.

3. The introduction refers to protocols that were published by the WHO, however the placement of reference 8 is misleading, as this is a review paper and not a WHO publication. This should be rephrased to clearly identify what work the reference alludes to.

Response: The reference of the review paper has been replaced with the WHO protocols webpage. In addition, references of several recently developed protocols have been added.

4. The author could explain more clearly why the presence of a proofreading exonuclease is relevant (i.e. it would theoretically reduce errors during replication/ enhance CoV replication fidelity).

Response: Part on exonuclease activity of Coronaviruses has been expanded along with specific mention of nsp14-ExoN in SARS-CoV-2.

5. The objective could be rephrased to "... re-assessment of the currently recommended PCR primers/probes for COVID-19 diagnostics through the evaluation of ..."

Response: The objective has been rephrased as suggested.

6. The manuscript would benefit from English language edits, particularly grammar. Organisation names should be checked (e.g. US Centers for Disease Control is incorrectly referred to as CDC USA). There is also inconsistent and sometimes inappropriate use of upper versus lower case for virus names.

Response: The names of organizations have been updated. Names of all the seven organizations are mentioned in the Methods section. Previously names of two organizations were mentioned in the Introduction that have been removed to avoid redundancy and geographical bias. The manuscript has been carefully proofread and the viral names have also been corrected.

7. Some references are missing information [23] and/or require reformatting [29].

Response: The references have been corrected (New reference number 34 & 37). One of the articles was an accepted version before publication and now a DOI has been added.

Reviewer: 3

Comments to the Author(s)

However, the proposal and methodology are still quite general. It could be further improved by clearer addressing the sub-questions intended to answer. For example, just some of my thoughts, one of the questions could be how the primers available now have mismatches with other viruses, such as influenza, etc. Another one could be investigating primer efficiency cross different SARS-CoV-2 lines as several mutations have been reported recently even within the infectious domains. Finally, it would also be interesting to line the results with the diagnostic situation among different countries.

Response: The objective and the question being asked has been further clarified in the Introduction section. In addition, the geographical analysis would be performed for the sequence variants with high frequency. The Issues with primer/probe mismatch for several other viruses including influenza virus has been discussed with references in the Introduction section (2nd paragraph of Introduction).

Overall, the author proposed a much-needed evaluation in the field, and the methods proposed are reasonably quick to do. But, before starting, the proposal would benefit from more detailed planning of what questions to answer.

Response: The objective and the questions being asked have been further clarified in the last paragraph of the Introduction section.

Reviewer: 4

This is a valid study, and while the results of this analysis would be of interest, another study has already demonstrated that mutations exist in SARS-CoV-2 at primer/probe binding regions (<https://www.medrxiv.org/content/10.1101/2020.03.30.20048108v1.full.pdf>). This paper would need to be addressed in the discussion.

Response: See our response below.

In regard to scientific validity, the research question in Table 1 about the existence of mutations in PCR primers/probes for SARS-CoV-2 has already been answered in the above paper. Nonetheless, that paper only scanned 992 genomes, and there would likely be benefit to doing a larger scale scan as proposed by this paper.

Response: See our response below.

The hypothesis stated is that mutations can cause mismatches in the primer/probe region. This hypothesis is logical but again has already been investigated in the above study.

Response: The reviewer shared a recent preprint by Vogels et. al. that addresses the same question as our study using 992 genome sequences and found a couple of mismatches. This timely study highlights the importance of studying the variability in the primer binding regions of PCR diagnostic assay. They have included 9 primer sets previously published by WHO and most of those were developed in the early phase of the pandemic when limited genomic information was available. In the meantime, more assays have been developed and published in the literature and we would include at least 20 assays in our study. We agree with the reviewer that larger scale scan of viral genomes would be useful to validate the findings of this study. Although the final number of genomes may vary depending on the inclusion/exclusion criteria mentioned in the manuscript, we plan to include >10,000 publicly available genomic sequences. Although the detailed methodology of the study by Vogels et. al. is not available, we plan to make our sequence files available so that any researcher may verify the variability in the region of their interest quickly.

As the preprints can be updated by the authors and would potentially be revised during the peer-review, we would cite and discuss the up-to-date (and possibly published) version of this important study in our manuscript at Stage 2 submission.

The analyses proposed are methodologically sound and feasible. They follow a study published recently.

Response: We agree that the methodology is quite straight forward and has successfully been applied to influenza virus previously.

There could be additional clarity around the specific parameters to be used in the multiple sequence alignment. Specifying these parameters would be important for reproducibility.

Response: Further details about methodology including multiple sequence alignment have been added in the pipeline summary (Fig 1).

Specialist editor comments:

Comments to the Author(s)

I think the rationale for this research is clear and I agree that the information is valuable, no matter what outcome will be presented.

My major concern is that while the authors specify that they plan to use sequences from a number of countries, I didn't see a proper interpretation of the information in the text. Let me explain what I mean: On one hand, the authors plan to use sequences originating from patients in different countries (which I think is good). On the other hand, they plan to use only high quality complete sequences (which again is good). I am not sure that these two statements necessarily overlap, and I am a little worried that this analysis could be biased based on that. I think that the authors should clearly address the geographical distribution and the potential biases.

Response: We agree with the concern and thus have added a paragraph in the Methods section regarding possible sources of geographical bias/data skew. These points would be further discussed in the discussion section during Stage 2 to ensure that valid conclusions are drawn from the data in term of geographical correlation.

Appendix B

Toronto, May 18, 2020

Professor Chris Chambers
Subject Editor, Registered Reports
Royal Society Open Science

FACULTY OF
SCIENCE

Department of
Biology

4700 KEELE ST
TORONTO ON
CANADA M3J 1P3

Tel 416 736 2100
Ext 77696
Cell 647 860 6799
kakhan@yorku.ca

Subject: COVID-19 Registered Report- Stage 2

Dear Professor Chambers,

I am submitting herewith a Stage 2 manuscript entitled “**Inclusivity analysis of diagnostic PCR assays through evaluation of sequence variation in coronavirus SARS-CoV-2 genome**” for consideration for publication in Royal Society Open Science. The URL for the accepted Stage 1 protocol registered on the Open Science Framework is provided in Methods section on page number 2 of the manuscript file.

This study conducted an inclusivity analysis of 23 published diagnostic PCR assays including the ones recommended by the World Health Organization (WHO). Exhaustive evaluation of the sequence variability within the primers/probe target regions of the viral genome performed using >17000 publicly available viral sequences from around the world showed mismatches in 6 assays. A comprehensive bioinformatics approach for *in silico* reassessment of PCR diagnostic assays of SARS-CoV-2 was validated using freely available software programs that can be applied to any diagnostic assay of choice.

To complete the requirement of submission, it is further confirmed that;

- No data for the pre-registered study was collected prior to the date of in principle acceptance (IPA).
- the completed analysis has been executed in the manner originally approved or as per deviation approved after Stage 1 (footnote was added for each change in the protocol).
- the introduction, the stated hypotheses, and the methods were not altered except for language changes (using track changes function). The title was changed to better reflect the study. Please let me know if it is not allowed.
- Raw and processed data (which is not available on a public archive or in the article) has been submitted as electronic supplementary material or has been deposited to OSF (see Data Accessibility section for details).

I believe that it would be a timely study keeping in view the current COVID-19 pandemic and would potentially provide important information for clinicians, laboratory professionals and decision-makers.

Yours sincerely,

Kashif Aziz Khan, DVM, PhD

Appendix C

Associate Editor Comments to Author (Professor Chris Chambers):

Associate Editor: 1

Comments to the Author:

Two of the original Stage 1 reviewers have now reappraised the Stage 2 manuscript, in addition to a specialist editor on the Biochemistry, Cellular and Molecular Biology editorial board, and the Registered Reports editor.

Both reviewers are broadly satisfied with the Stage 2 submission while also noting several areas requiring clarification. Reviewer 1 notes the need for a clearer definition and justification of the term "inclusivity analysis" (especially given that the term is now in the title), and both reviewers question the rationale for the 0.5% threshold, which appears to have been decided after Stage 1 acceptance. The selection of this threshold -- and the point in time that it was decided (before or after observation of the data) -- needs to be made clear.

Response:

Please see the response to the reviewers below for a detailed reply to both concerns mentioned above.

In addition to the reviews, I have passed through the manuscript and made detailed suggestions for edits to improve grammar, clarity and presentation. Please find attached an annotated PDF with these additional comments and suggestions.

Response:

- *Most of the minor suggestions have been included.*
- *The abstract and the conclusion have been revised to answer the original question "Are there any mutations in the primer/probe binding regions of the SARS-CoV-2 genome for PCR assays published in the literature?"*
- *The concern of skew due to the data exclusion has been addressed by analyzing the data from the 3 primers/probe with the highest number of "removed" sequences (see new electronic supplementary material, file 4). The excluded data for all the primers/probes was not analyzed as we are not making any claims about geographical correlation but we want to provide the geographical analysis for the readers who might want to draw some conclusions for a specific geographical location.*

As a side note, data for 4 additional assays are included in the tables, supplementary figure and throughout in the text. Certain mistakes were corrected in table 2.

Reviewer: 1

Comments to the Author(s)

The Introduction, rationale and stated hypotheses are the same as those approved at Stage 1 submission. However, the title has changed. The term "Inclusivity analysis" is a bit confusing, both in the title and throughout the paper. It should be properly defined in the text, and ideally a reference provided. I assume that it means something along the lines of 'proportion of the sequenced virus isolates that would be expected to be detected by a given assay, based on perfect matches in the primer/probe binding sequences'?

Response: *As the term inclusivity analysis to define match/mismatches was introduced in stage 2, it was causing some confusion. So, we stepped back and removed it from the title and term mismatches was used in the title and throughout the article as it was used in the original hypothesis (see table 1). Now the inclusivity is mentioned only once in the abstract and defined in the discussion section along with the references.*

The experimental procedures were carried out as planned. The only exception was that base composition of each nucleotide position was not analyzed, although this was justified based on their findings of moderate sequence variation. Additional methods details are provided in comparison to the original proposal. These are presented as part of the "Results" section, where they more clearly outline some of the steps taken. These are still in line with the original proposal. The one potential exception is the introduction of a threshold for inclusivity analysis. This concept could be explained in slightly more detail. I assume the authors mean that a variant was only considered if it was present in > 0.5% of the informative aligned sequences (i.e. > ~85 isolates, depending on the MSA)? This seems quite lenient, i.e. you might miss important variants, particularly if the same mutations are found from isolates sequenced independently in different labs. However, I think it is fine to keep this threshold, but providing a bit more discussion on this subject, and defining the threshold more clearly. This concept is generally clear from the Tables and Figures, but less so from the text.

The author's conclusions are justified given the data.

Response: *The introduction of the threshold was decided before performing the study and the reference to the previous relevant study used was provided in stage 1 protocol, but the threshold was not specifically mentioned. It was partially due to the lack of our experience with the "registered report" format. Now it has been defined well in Methods and then explained further in the first paragraph of Results.*

Indeed, the threshold was defined as $\geq 0.5\%$ of all the sequences rather than informative sequences. This ensures the same threshold for all the diagnostic assays and avoids the effect of removed data on the threshold.

Suggested minor edits:

In general, the manuscript would benefit from English language edits, particularly the most recently added sections. Data should be referred to as plural throughout. In many cases, this is also true for 'assay'.

Summary:

Methods:

Discussion:

Response: *Modified as suggested.*

Results

-4.1 Lines 32-39: This is more appropriate material for the Introduction, and does not represent results of this study. Suggest either moving to Introduction or removing completely.

Response: *Moved to the Introduction as suggested.*

-4.1 Lines 41-47: This would be more appropriate in the Methods section.

Response: *Removed from the Result with partial modification of Methods to include the specific number of sequences included.*

-4.2 p3 Lines 49- p4 Line 12: This would also be more appropriate in the Methods section.

Response: *This portion has partially been kept because minimal modification of Methods is allowed at this stage. However, it has been modified to include the rationale of the threshold.*

-p3 Line 18 – It's not quite sure what is meant by "one mismatch with all the sequences above threshold". Perhaps the terms "above threshold" and "informative sequences" are being mixed?

Response: *To be consistent, mismatches were mentioned from the sequences above the threshold. In some cases, including this one, all the informative sequences showed a mismatch. Indeed, there was a mistake in the sequence tracing figure that has been corrected and the words "above threshold" has been removed from the text.*

Figures:

Tables:

Supp File 2

Response: *Modified as suggested.*

References

-Some references are missing information [5, 12, 14, etc.] and should be checked for proper formatting.

Response: *The information is missing for some references as those are early online version ahead of publication. So DOI only is available. The References have been updated where information became available in the meantime.*

Reviewer: 3

Comments to the Author(s)

The paper is well-organised and I am glad to see that the author has addressed my concerns. The introduction, rationale and hypotheses are the same as stage1. The results and conclusions are sound.

However, the result parts were a bit wonky and detailed descriptions of analysed data were missing, such as the pie charts provided on page 20.

Response: *The description of the analyzed data about pie charts has been expanded in Results.*

Also, the reason under the chosen threshold was firstly described in the discussion part, which needs to be carefully explained before in the result part, along with how does threshold change affect the whole outcome need to be explained.

Response: *Threshold has been defined well in Methods and then explained further in the first paragraph of Results now. The possibility of a lower threshold along with possible outcomes has been addressed in Discussion.*